# Use of Tandem Mass Spectrometry Quantitative Proteomics to Identify Potential Biomarkers to Follow the Effects of Cold and Frozen Storage of Muscle Tissue of *Litopenaeus vannamei*

**DOI:** 10.3390/foods12152920

**Published:** 2023-07-31

**Authors:** Yu Chen, Qian Ning, Zhenzhen Wu, Hanlin Zhou, Jun Liao, Xiangyun Sun, Jing Lin, Jie Pang

**Affiliations:** 1College of Food Scientific, Fujian Agriculture and Forestry University, Fuzhou 350002, China; yuchen_0520@163.com (Y.C.); wuzhenzhen@fafu.edu.cn (Z.W.); zhouhanlin@fafu.edu.cn (H.Z.); liaojun1210@163.com (J.L.); sunxiangyun5723@163.com (X.S.); linjing5577@163.com (J.L.); 2Jinshan College of Fujian Agriculture and Forestry University, Fuzhou 350001, China; qianqian1000@163.com

**Keywords:** TMT, proteomics, *Litopenaeus vannamei*, low-temperature storage

## Abstract

*L. vannamei* has become one of the most productive species. However, it is susceptible to microbial contamination during fishing, transportation, and storage, which can lead to spoilage and quality deterioration. This study investigates the relationship between changes in the proteome of *Litopenaeus vannamei* (*L. vannamei*) muscle and quality characteristics during low-temperature storage using the tandem mass spectrometry technology of quantitative proteomics strategy. The differential expression of proteins under cold storage (4 °C, CS), partial slight freezing (−3 °C, PFS), and frozen storage (−18 °C, FS) conditions was compared with the fresh group (CK), resulting in 1572 proteins identified as differentially expressed. The purpose of this research is to identify potential biochemical markers by analyzing quality changes and relative differential proteins through searches in the UniProt database, Gene Ontology database, and Genome Encyclopedia. Correlation analysis revealed that seven DEPs were significantly related to physical and chemical indicators. Bioinformatics analysis demonstrated that most DEPs are involved in binding proteins, metabolic enzymes, and protein turnover. Additionally, some DEPs were identified as potential biomarkers for muscle decline. These findings contribute to understanding the mechanism of freshness decline in *L. vannamei* under low-temperature storage and the changes in muscle proteome.

## 1. Introduction

*Litopenaeus vannamei* (*L. vannamei)* is a highly sought-after shrimp species in China due to its transparent shell, delicious meat, low fat content, and rich nutritional value [1]. With the increasing demand for high-nutrient shrimps, *L. vannamei* has become one of the most productive species internationally [2,3]. However, due to its high protein and moisture content, and the presence of polyphenol oxidase, *L. vannamei* is susceptible to microbial contamination during fishing, transportation, and storage, which can lead to spoilage and quality deterioration [4,5]. The fresh-keeping problem is further compounded by the numerous processes involved, including factory processing, warehouse storage, and transportation [6,7], resulting in changes in various freshness indicators [8]. Low-temperature storage is the most common method used to preserve the freshness and quality of *L. vannamei*.

Low-temperature storage effectively inhibits the growth and reproduction of microorganisms and the activity of polyphenol oxidase in shrimps, thereby extending their shelf life. However, cold treatment can result in the loss of moisture and recrystallization, leading to low water holding capacity, protein denaturation, and fat oxidation, which significantly reduce the edible value of *L. vannamei* [9], which refers to the size and the number of crystals which are closely dependent on the temperature of storage [10].

Proteomics, as a research hotspot in life sciences, is used to explore the relationship between muscle protein and quality traits at the molecular level. It has a strong role in revealing the biological processes, molecular functions, and cell components that affect quality traits at the protein level [11,12]. Previous studies have used two-dimensional gel electrophoresis (2-DE) and high-resolution liquid chromatography–tandem mass spectrometry (LC-MS/MS) to study the relationship between muscle protein and different quality traits. DE proteomics and multivariate data analysis methods were employed to study the changes in cod (Gadus morhua) muscle protein during cold storage [6]. Li et al. found five spots of interest in the 2-DE map of large yellow croaker under cold storage conditions and proposed that three structural proteins and two metabolic proteins may be potential freshness markers [13]. However, the molecular weight and properties of the protein, such as being too large or too small, having low hydrophobicity, or being extremely acidic or alkaline, can make separation and identification difficult [10,14].

In recent years, high-resolution mass spectrometry technology has become a developed technology in global proteomics technology. Isobaric iTRAQ for relative and absolute quantification, a relative quantification technique for in vitro equivalent isotope labeling, can compare the protein expression of four to eight samples at the same time. Duan et al. reported that different types of microplastics (MPs) induce different changes in the hemolymph protein profile and identified several stress-related differentially expressed proteins (DEPs) and metabolite markers [15]. The toxicology of five MPs to Lactobacillus vannamei affects the DEPs and metabolites of *L. vannamei* [14]. For the first time, proteins that may interact with hemocyanin were reported, and 39 possible hemocyanin proteins were identified, including the coagulation-related factor transglutaminase [11,16]. Compared with iTRAQ, tandem mass spectrometry (TMT) combined with high-resolution mass spectrometer technology can simultaneously identify and quantify proteins from three different samples, realize high-throughput proteomics quantitative analysis, and reduce experimental errors [17]. However, the use of TMT technology for proteomic determination and bioinformatics combined with physical and chemical comprehensive analysis of *L. vannamei* during low-temperature storage has been rarely reported.

Previous studies have also shown that the ribosomal protein content of frozen pipe whip prawns decreased, and the ribosomal protein content was significantly positively correlated with springiness and chewiness [10]. 40S ribosomal protein S3a, 60S ribosomal protein L18a, and ribosomal proteins L3, L8, and L18 were all downregulated in frozen white-leg shrimp [18]. CCT-epsilon is related to protein folding [19], and kinases are involved in protein phosphorylation [20], both of which play an important role in protein synthesis. Therefore, the downregulation of CCT-epsilon and kinase content may result in the downregulation of protein content during storage.

In this study, we utilized a combination of TMT and LC-MS/MS techniques to investigate the impact of three different storage temperatures (4, −3, and −18 °C) on *L. vannamei* muscle over several days. The aim was to identify potential biochemical markers by analyzing relative differential proteins associated with changes in quality. Protein analysis will shed light on the molecular mechanisms underlying *L. vannamei* quality deterioration and offer novel insights into *L. vannamei* muscle subjected to various cold storage treatments.

## 2. Materials and Methods

### 2.1. Chemical Reagents

Chloroform methanol, acetonitrile, isopropanol, and acetonitrile (≥99.9%) were purchased from Thermo Fisher Scientific (Waltham, MA, USA). Ammonium formate (≥99.9%) was purchased from Sigma-Aldrich (Shanghai, China). All other reagents commonly used for lipid extraction were obtained from Tedia Company Inc. (Fairfield, OH, USA).

### 2.2. Sample Preparation

Fresh *L. vannamei* (10–12 cm in length) was purchased from Yonghui Supermarket in Fuzhou City and transported on ice for 1 h. The specimens were then stored in a Midea freezer 247 L (BCD-247WTM). To create three groups of *L. vannamei* of equal size, they were randomly divided and placed into Ziplock bags with 8–10 *L. vannamei* in each. These bags were stored at three different temperatures: 4 °C (cold storage, CS), −3 °C (partial slight frozen, PSF), and −18 °C (frozen storage, FS) for 10, 30, and 60 days, respectively. After different time periods of storage, the samples were taken out, and the *L. vannamei* heads and shells were removed and crushed. The muscle samples used for proteomics analysis were immediately placed in nitrogen and stored in a freezer at −80 °C until use. The chemical and physical indicators of the remaining muscles were measured.

### 2.3. Quality Determination

To indicate the freshness of the shrimp, we measured various indicators, including water holding capacity, color, texture, and total volatile basic nitrogen (TVB-N). Water holding capacity was measured following the method proposed by Priyadarshini et al. [21]. Color characteristics, including the lightness (L*) of the shrimp meat, were determined using a colorimeter (ADCI-60, TAINA, Shanghai, China) as described by Yuan et al. (2016) [7]. Texture analysis involved measuring various parameters such as hardness, springiness, chewiness, cohesiveness, and shear force using a texture analyzer (TA) (XT Plus, Stable Microsystems, UK) and blade set (HDP/UK), with the test conditions set as follows: the second segment after color measurement was used to determine the aforementioned parameters using the second extrusion method of the texture analyzer. The probe used was a P/50 flat-bottomed cylindrical probe with a diameter of 50 mm, the test rate was 1 mm/s, and the compression rate was 50%. Shear force was determined at a continuous speed of 2.0 mm/s using the blade set (HDP/UK), with the main peak recorded as the first record [22]. Volatile basic nitrogen (TVB-N) was determined according to the method described in GB 5009.228-2016 “Determination of Volatile Base Nitrogen in Food” [23]. All results were expressed as the mean ± standard deviation.

### 2.4. Protein Extraction and Peptide Digestion

The samples were lysed using SDT Lysis Buffer containing 4% (*w*/*v*) Sodium Dodecyl Sulfonate (SDS), 100 mM Tris/HCl pH 7.6, and 0.1 M Dithiothreitol (DTT) to extract protein. Protein quantification was performed using the Bicinchoninic Acid (BCA) method. An appropriate amount of protein was taken from each sample and subjected to Trypsin digestion using the Filter-Aided Proteome Preparation (FASP) method [24]. The peptides were desalted using a C18 Cartridge (2.1 × 100 mm, 1.7 µm, Waters, MA, USA), freeze-dried, and reconstituted in 40 μL of 0.1% formic acid solution for peptide quantification (OD280).

### 2.5. TMT Labeling of Peptide and Peptide Classification

Labeling of 100 μg peptides from each sample was carried out according to the instructions of Thermo Company’s TMT labeling kit (Thermo Fisher Scientific, USA). There were 4 design groups, each containing 2 biological replicate samples. Labeled peptides were mixed for each group and classified using an automatic protein chromatography system (AKTA Purifier 100). Buffer A solution was 10 mM KH_2_PO_4_, 25% CAN, pH 3.0, and buffer B solution was 10 mM KH_2_PO_4_, 500 mM KCl, 25% CAN, pH 3.0. The chromatographic column was balanced with liquid A, and the sample was loaded from the injector to the chromatographic column for separation at a flow rate of 1 mL/min.

The liquid gradient was as follows: 0% B liquid for 25 min, B liquid linear gradient from 0 to 10% for 25–32 min, B liquid linear gradient from 10 to 20% for 32–42 min, B liquid linear gradient from 20 to 45% for 42–47 min, the linear gradient of B solution from 45 to 100% for 47–52 min, B solution maintained at 100% for 52–60 min, and B solution reset to 0% after 60 min. During the elution process, the absorbance value at 214 nm was monitored, and the eluted fractions were collected every 1 min and then desalted using a C18 Cartridge after lyophilization.

### 2.6. UPLC-QE-MS/MS Analysis

For data collection, each sample was separated using an HPLC liquid system, Easy nLC (Thermo Scientific, Pleasanton, CA, USA), with a nanoliter flow rate. Buffering solution A was a 0.1% formic acid aqueous solution, while solution B was a 0.1% formic acid acetonitrile aqueous solution (acetonitrile content: 84%). The chromatographic column was equilibrated with 95% solution A, and the sample was loaded by the autosampler onto the loading column (Thermo Scientific Acclaim PepMap100, 100 μm × 2 cm, nanoViper C18) and then passed through the analytical column (Thermo Scientific EASY column, 10 cm, ID 75 μm, 3 μm, C18-A2) for separation, with a flow rate of 300 nL/min. After chromatographic separation, the samples were analyzed by a Q-Exactive mass spectrometer. The detection method was positive ion, and the scanning range of precursor ions was 300–1800 m/z. The resolution of the primary mass spectrum was 70,000 at 200 *m*/*z*, with an AGC (automatic gain control) target of 1 × 10^6^, and a dynamic exclusion time of 60.0 s. The mass-to-charge ratios of peptides and peptide fragments were collected using the following method: 20 fragment maps (MS2 scans) were collected after each full scan. The Mass Spectrometry2 (MS2) activation type was HCD, and the isolation window was 2 *m*/*z*. The MS2 resolution rate was 17,500 at 200 m/z, with a normalized collision energy of 30 eV and an underfill of 0.1%.

### 2.7. Protein Database Searching and Analysis

Raw data from LC-MS/MS were analyzed using MaxQuant software (v1.6.2.10). Data interpretation and protein identification were performed against the transcriptome database. The parameter settings included strict trypsin specificity, a maximum of two missed cleavages, carbamidomethyl as a fixed modification, and oxidation as variable modifications. The database for *L. vannamei* was obtained from UniProt (http://www.uniprot.org/, accessed on 1 January 2022). Mass tolerance of the first-stage precursor was set at 20 ppm, and 0.1 Da for the second fragment ions. The first discovery rate (FDR) of protein identifications was set at 1% using the TMT-6/10/16plex approach, and at least one unique peptide was identified for each protein. The protein quantification method involved calculating the protein ratio as the median value of the unique peptide of the protein.

### 2.8. Bioinformatics and Statistical Analysis

The UniProt-Decapoda 20180820 database was used for retrieval and quantitative analysis. Proteome Discoverer 1.4 was used as the search database software to perform differential protein screening using the condition of 1.2-fold change (FC) and *p* < 0.05. An FC ≥ 1.2 indicates upregulation, FC ≤ 0.83 indicates downregulation, and 0.83 < FC < 1.2 indicates no obvious change. The target protein set was annotated using Blast2GO, the KEGG pathway was annotated using KAAS (KEGG Automatic Annotation Server) software (http://geneontology.org/, accessed on 22 June 2023), and Fisher’s exact test was used to compare each KEGG classification. KEGG annotation enrichment analysis was conducted on the target protein collection and the distribution of the overall protein collection was analyzed. Pearson’s correlation analysis of DEPs with quality characteristics (water holding capacity, L*, hardness, springiness, chewiness, cohesiveness, and TVB-N) of *L. vannamei* muscles was performed using SPSS 19.0. Adjusted *p*-values were derived from multiple statistical tests to correct for the occurrence of false positives. FDR (false discovery rate) is a common term in statistics. Its meaning is the expected value of the proportion of the number of false rejections to the number of all rejected original hypotheses.

## 3. Results and Discussion

### 3.1. Quality Analysis of L. vannamei Muscles

Water holding capacity, color, texture, and TVB-N are common physical and chemical indicators used to evaluate the freshness of aquatic products such as shrimps. Table 1 shows the changes in muscle quality characteristics of *L. vannamei*. The water holding capacity of shrimp decreased with the increase in storage time at three different temperatures compared to the CK group. The water holding capacity increased when thawed and heated, resulting in hard and fragile meat. The increase in water holding capacity reduces the free water on the surface of the shrimp meat, enhancing the reflection of light, resulting in gradual whitening of the shrimp meat and higher L*. The indicators of texture characteristics (hardness, springiness, chewiness, cohesiveness) decreased after storage at 4 °C for 10 days, storage at −3 °C for 30 days, and storage at −18 °C for 60 days compared to the CK group. At lower storage temperatures, the activity of enzymes and microorganisms is weaker due to lower protein degradation, resulting in smaller texture changes [25].

During storage, the increase in the content of α-type proteasome promotes the decomposition of damaged proteins, resulting in an increase in TVB-N content [26,27]. The results of correlation analysis showed that α-type proteasomes had a very significant positive correlation with TVB-N content. The TVB-N content increases with the decomposition of protein in shrimp meat. Therefore, it is necessary to evaluate the freshness comprehensively by judging from multiple indicators [25].

TVB-N is an important physical and chemical index used to measure the freshness of protein foods such as fish, shrimp, and shellfish and the quality of aquatic products. The water holding capacity of aquatic products refers to the ability of the muscles to retain their own water and additional water during processing. With the increase in storage time, the water holding capacity of *L. vannamei* decreases. When thawing and heating, the loss of juice increases, causing the meat to become hard and fragile. Therefore, water holding capacity is also an important quality indicator to measure the freshness of meat [28]. In addition, color can also reflect the freshness of *L. vannamei*. As water holding capacity increases, more water is free on the surface of the meat, which enhances the reflection of light, resulting in whiter meat and higher L* [25].

Furthermore, the texture characteristics of aquatic products are important indicators for evaluating their freshness, which can reduce evaluation errors caused by subjective human factors [25].

### 3.2. Protein Identification and Statistical Analysis

The peptides obtained from protein digestion were labeled, graded, and analyzed using mass spectrometry. The analysis revealed a total of 11,326 peptides, of which 9071 were specific polypeptides. The total number of identified protein groups was 1610, including 1572 quantifiable proteins. Interestingly, 68.32% of the identified proteins contained at least two peptides (Figure 1A). Furthermore, the identified proteins had high sequence coverage, with 48.88% of protein sequence coverage exceeding 10% (Figure 1B). Proteins with a molecular weight of less than 60 kDa accounted for 69.13% (Figure 1C). The length distribution of identified peptides showed a unimodal distribution, mostly composed of 5–15 amino acids, with peaks for peptides with a length of 7–9 amino acids (Figure 1D).

In comparison to label-free proteomics technology, TMT quantitative proteomics identified 612 proteins in shrimp muscle, including 240 differentially abundant proteins (DAPs) [29]. However, our study identified a much higher number of specific peptides and proteins using TMT quantitative proteomics technology. These results indicate that the TMT quantitative proteomics technology has high coverage and sensitivity, and the identified proteins are comprehensive and reliable.

### 3.3. Statistics Analysis of Differential Proteins

Differential proteins were categorized based on proteins with FC > 1.2 and *p* < 0.05. In comparison with the CK group, a total of 181 proteins showed significant changes in different preservation temperatures, with 84 upregulated and 97 downregulated proteins (Table 2 and Figure 2). Furthermore, a total of 111 differential proteins were identified in the CS/CK comparison group, which was considerably higher than the other two groups, indicating that relatively higher storage temperature caused more significant changes in shrimp protein.

Regarding common differential proteins, three, one, and four proteins were upregulated in the CS and PFS, CS and FS, and FS and PFS groups, respectively. However, there were 17, 1, and 3 common downregulated proteins in the same groups.

### 3.4. Functional Analysis of Differential Proteins

Differential proteins were analyzed based on their Gene Ontology (GO) functional classifications, which are internationally standardized classifications of gene functions [30]. The GO analysis in this study comprised three important subjects of biological function, namely, biological process (BP), molecular function (MF), and cellular component (CC). As depicted in Figure 3 and Table 3, the highest expression categories in BP for the CS/CK, PFS/CK, and FS/CK comparison groups were cellular processes, metabolic processes, and biological regulation processes. In MF, the highest expression categories were binding and catalytic activity, while in CC, the highest expression categories were cell parts and organelles. These results are similar to the study during storage by Roncaglia et al. (2013) [31].

To reveal the overall metabolic pathway enrichment characteristics of all differentially expressed proteins, Fisher’s exact test was performed for KEGG pathway enrichment analysis. The top ten pathways with different protein enrichment factors were shown in the KEGG pathway enrichment bubble chart (Figure 4). In the CS/CK comparison group, ketone body synthesis and degradation, Cyano group acid metabolism, and apoptosis pathways changed, with five differential proteins involved in the ubiquitin-mediated proteolysis pathway. In the PFS/CK comparison group, cell adhesion molecules, binary system, and nucleotide excision repair pathways changed, and four different proteins were involved in the oxidative phosphorylation process. In the FS/CK comparison group, the pathways of apoptosis, limonene and pinene degradation, and hormone biosynthesis changed, with four differential proteins involved in the oxidative phosphorylation process.

### 3.5. Correlation Analysis between Differential Protein and Quality Index

This study selected 11 representative quality indicators, including water holding capacity, L*, hardness, springiness, chewiness, cohesiveness, and TVB-N, to explore the correlation between different proteins and various quality indicators. Through correlation analysis, a total of 109 differential proteins were identified in the three comparison groups of CS/CK, PFS/CK, and FS/CK, reaching significant differences in at least one of the comparison groups. The contents of these differential proteins were then correlated with 11 representative indicators.

The correlation analysis results showed that 21 differential proteins were extremely significantly related to changes in at least one of the seven quality indicators. Among the seven different proteins involved in protein biosynthesis, all were positively correlated with springiness, cohesiveness, and water holding capacity but negatively correlated with hardness and L*. Of these, three ribosomal proteins (40S ribosomal protein S8/40S ribosomal protein S27/S10e ribosomal protein) and Mitogen-activated protein kinase 7-interacting protein were significantly or extremely significantly positively correlated with chewiness and were significantly or extremely significantly negatively correlated with L*. Only CCT-epsilon protein had a very significant positive correlation with springiness and a significant negative correlation with hardness. In addition, the content of three ribosomal proteins was significantly or extremely significantly positively correlated with water holding capacity.

The two different protein contents involved in protein degradation showed the opposite trend to the protein biosynthesis process. Among them, the proteasome subunit alpha type had a very significant positive correlation with TVB-N content, and the peptide hydrolase was significantly related to springiness and chewiness or had a very significant negative correlation.

Two proteins, putative RNA-binding protein-like isoform X4 and sex-lethal, were involved in the RNA binding process and were both negatively correlated with L* and positively correlated with cohesiveness and chewiness. The correlation between the two proteins involved in the GTP binding and quality indicators showed an opposite trend. The J-domain-containing protein was significantly or positively correlated with chewiness and water holding capacity and significantly negatively correlated with L*. RhoA was significantly or significantly negatively correlated with cohesiveness and water holding capacity.

Other proteins included four that were not annotated in the biological process and molecular function of GO annotation. Putative dysferlin-like protein had a very significant negative correlation with water holding capacity and chewiness and a significant positive correlation with L*. The remaining three proteins (troponin i/activation factor subunit spp27/putative ensconsin-like isoform X8) were significantly or extremely significantly correlated with chewiness and water holding capacity, with a positive correlation and negative correlation with L*. In addition, the four unannotated proteins all had a significant or very significant positive correlation with chewiness and a very significant negative correlation with L*. The two protein contents of A0A423SR08 and A0A3R7PV59 were significantly positively correlated with water holding capacity.

Overall, the correlation analysis provided insights into the relationship between differential proteins and various quality indicators, which could contribute to a better understanding of the mechanisms underlying meat quality.

## 4. Conclusions

In this study, we utilized TMT quantitative proteomics technology to investigate differential protein expression in *L. vannamei* stored under different methods. Our results showed that troponin content was downregulated during storage and was significantly correlated with chewiness and cohesion. This was likely due to ice crystals destroying the muscle structure of shrimp meat, which reduces the water wrapped by myofibrils and reduces water retention, leading to protein denaturation and degradation in the muscle tissue.

Moreover, troponin content was significantly correlated with four quality indicators, indicating its potential as an indicator protein for monitoring quality changes in *L. vannamei* during storage. Our results also identified four unannotated proteins that exhibited extremely significant negative correlations with L* and chewiness, highlighting their potential as targets for further investigation of their functions.

In conclusion, our study sheds light on the mechanisms underlying protein degradation during *L. vannamei* storage and provides insights into potential indicator proteins for monitoring quality changes. Our findings also emphasize the importance of proper storage methods in preserving shrimp meat quality. Further research is warranted to explore the functions of unannotated proteins and their potential use in improving shrimp storage and preservation. 

## Figures and Tables

**Figure 1 foods-12-02920-f001:**
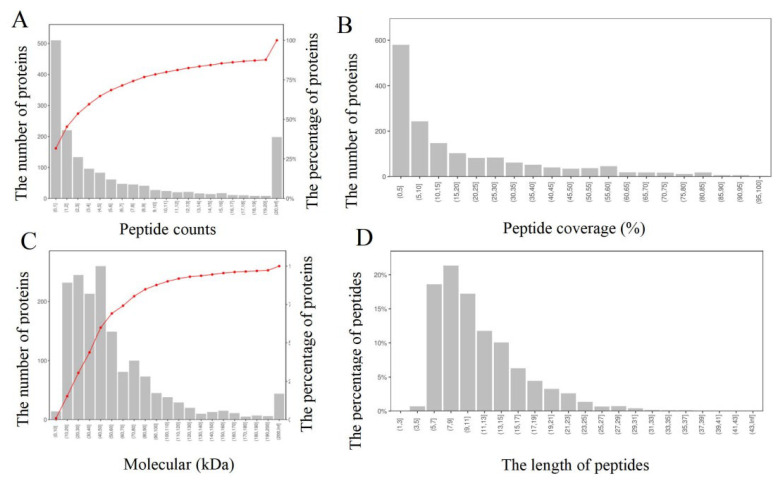
Distribution of protein peptide number (**A**), peptide coverage (**B**), protein molecular weight (**C**), and the length of peptides (**D**) in protein.

**Figure 2 foods-12-02920-f002:**
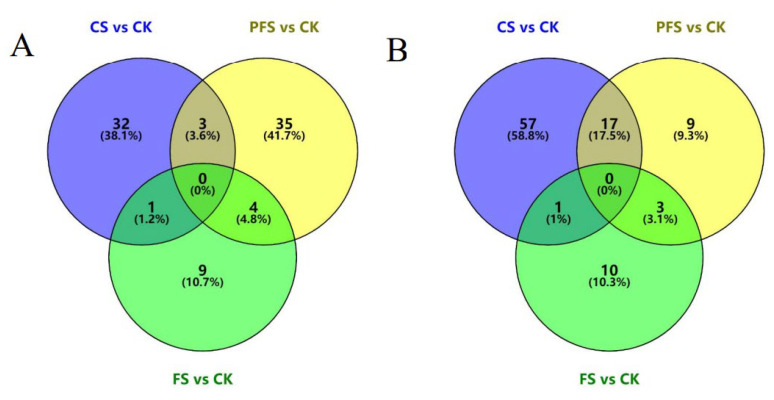
Venn diagram of upregulated proteins (**A**) and downregulated proteins (**B**) between *L. vannamei* under different storage methods and fresh ones.

**Figure 3 foods-12-02920-f003:**
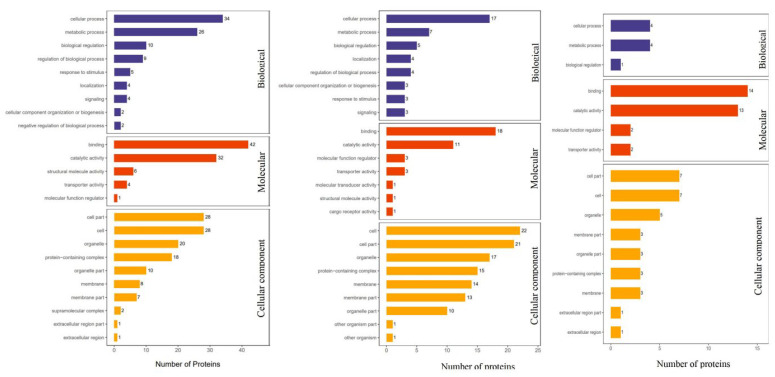
GO function annotation of differential proteins in different comparison groups of *L. vannamei*.

**Figure 4 foods-12-02920-f004:**
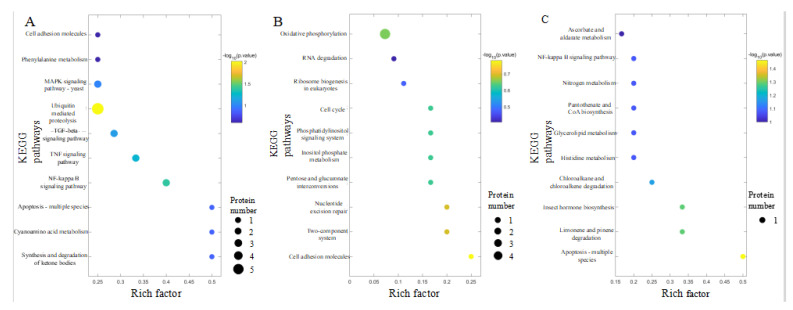
Enrichment analysis of differential protein KEGG pathway in each comparison group (top 10). (**A**) CS/CK; (**B**) PFS/CK; (**C**) FS/CK.

**Table 1 foods-12-02920-t001:** Quality characteristics of *L. vannamei* muscles in fresh group (CK) and three low-temperature treatments (CS, PFS, FS).

Quality Characteristics	q	CS	PFS	FS
Water holding capacity	90.83 ± 0.06 ^a^	81.43 ± 1.38 ^c^	85.31 ± 0.81 ^b^	88.38 ± 1.2 ^a^
L*	29.70 ± 0.41 ^a^	41.12 ± 0.32 ^a^	39.66 ± 0.15 ^b^	31.48 ± 0.18 ^c^
Springiness (N)	0.93 ± 0.00 ^a^	0.68 ± 0.01 ^c^	0.95 ± 0.00	0.98 ± 0.00 ^a^
Cohesiveness	0.44 ± 0.01 ^a^	0.33 ± 0.01 ^c^	0.41 ± 0.01 ^b^	0.43 ± 0.02 ^a^
Chewiness (mJ)	752.92 ± 5.01 ^a^	703.21 ± 28.86 ^a^	591.42 ± 27.61 ^a^	609.95 ± 40.78 ^b^
Hardness (g)	1733.54 ± 35.17 ^a^	2120.72 ± 77.57 ^a^	1435.13 ± 16.87 ^b^	1204.87 ± 36.08 ^b^
TVB-N	6.57 ± 0.54 ^a^	22.11 ± 0.19 ^a^	18.57 ± 0.41 ^b^	19.11 ± 0.29 ^b^

Note: Values are mean ± standard deviation. “a–c” letters indicate significant differences (*p* < 0.05).

**Table 2 foods-12-02920-t002:** GO function annotation of differential proteins in different comparison groups of LNVN.

Comparisons		CS/CK	PFS/CK	FS/CK
BP	cellular processes	34	26	10
	metabolic processes	26	7	4
	biological regulation processes	10	5	1
MF	binding	42	18	14
	catalytic activity	32	11	13
CC	cells	28	22	7
	cell parts	28	21	7
	organelles	20	17	5

**Table 3 foods-12-02920-t003:** Correlation analysis of differential proteins and quality indices of *L. vannamei* under different storage methods.

Accession	Protein Name	Springiness	Cohesiveness	Chewiness	Hardness	*L**	*a**	*b**	Water Holding Capacity	TVB-N
Protein biosynthesis										
A0A3R7Q994	40S ribosomal protein S8	0.585	0.815	0.989 *	−0.465	−0.993 **	0.416	−0.472	0.946	−0.804
A0A3R7QE43	40S ribosomal protein S27	0.732	0.916	0.997 **	−0.624	−0.975 *	0.559	−0.275	0.994 **	−0.827
A0A423TWT2	S10e ribosomal protein	0.682	0.878	0.997 **	−0.575	−0.997 **	0.534	−0.400	0.969 *	−0.766
A0A076NBT3	60S ribosomal protein L40	0.906	0.993 **	0.921	−0.831	−0.868	0.761	0.028	0.974 *	−0.747
A0A423U985	Putative eukaryotic translation initiation factor 4-like	0.947	1 **	0.886	−0.887	−0.833	0.828	0.077	0.947	−0.67
A0A3R7P4W6	CCT-epsilon	0.994 **	0.967 *	0.747	−0.967 *	−0.678	0.915	0.284	0.842	−0.529
A0A3R7N5V9	Mitogen-activated protein kinase 7-interacting protein	0.569	0.797	0.981 *	−0.454	−0.996 **	0.421	−0.526	0.927	−0.752
Protein hydrolysis										
A0A423T969	Proteasome subunit alpha type	−0.471	−0.709	−0.855	0.333	0.801	−0.198	0.162	−0.868	0.995 **
A0A423SW39	Acyl-peptide hydrolase	−0.966 *	−0.996 **	−0.859	0.918	0.813	−0.876	−0.080	−0.92	0.593
RNA binding										
A0A3R7PPQ9	Putative RNA-binding protein-like isoform X4	0.603	0.819	0.984 *	−0.493	−0.999 **	0.465	−0.507	0.933	−0.732
A0A423TSY3	Sex-lethal	0.938	0.994 **	0.877	−0.873	−0.814	0.800	0.124	0.947	−0.712
Accession	Protein name	Springiness	Cohesiveness	Chewiness	Hardness	*L**	*a**	*b**	Water holding capacity	TVB−N
GTP binding										
A0A3R7MKG0	J domain-containing protein	0.757	0.922	0.993 **	−0.662	−0.985 *	0.623	−0.322	0.980 *	−0.736
A0A3R7QNM6	RhoA	−0.937	−0.996 **	−0.883	0.872	0.822	−0.801	−0.110	−0.950 *	0.711
Other proteins										
A0A423SU24	Putative dysferlin-like	−0.754	−0.929	−0.991 **	0.648	0.963 *	−0.577	0.226	−0.998 **	0.836
K7WFT8	Troponin I	0.832	0.967 *	0.978 *	−0.743	−0.950 *	0.690	−0.176	0.993 **	−0.755
A0A423TC50	Activation factor subunit spp27	0.83	0.966 *	0.963 *	−0.736	−0.918	0.657	−0.085	0.997 **	−0.822
A0A423U5T5	Putative ensconsin-like isoform X8	0.729	0.914	0.994 **	−0.62	−0.968 *	0.549	−0.254	0.995 **	−0.845
Uncharacterized protein										
A0A423SPE6	Uncharacterized protein	0.707	0.881	0.982 *	−0.614	−0.992 **	0.595	−0.421	0.948	−0.677
A0A423SR08	Uncharacterized protein	0.674	0.872	0.996 **	−0.568	−0.998 **	0.529	−0.414	0.965 *	−0.758
A0A3R7PAK4	Uncharacterized protein	0.604	0.811	0.973 *	−0.501	−0.996 **	0.485	−0.532	0.917	−0.683
A0A3R7PV59	Uncharacterized protein	0.644	0.858	0.997 **	−0.529	−0.992 **	0.473	−0.398	0.970 *	−0.818

**Note:** Values are mean ± standard deviation. Data on the same row with different superscripts indicate a significant difference (*p* < 0.05). * Significant correlation (*p* < 0.05). ** Highly significant correlation (*p* < 0.01).

## Data Availability

The data that support the findings of this study are available from the corresponding author upon reasonable request.

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
