# Peer review of "Use of Tandem Mass Spectrometry Quantitative Proteomics to Identify Potential Biomarkers to Follow the Effects of Cold and Frozen Storage of Muscle Tissue of Litopenaeus vannamei"

_foods, 2023, doi:10.3390/foods12152920_

Round 1
Reviewer 1 Report
In the present manuscript, possible proteins were investigated as potential biochemical markers responsive to changes in the proteome of Litopenaeus vannamei muscle (L. vannamei - (China shrimp species) as well as to quality characteristics during low temperature storage using tandem mass spectrometry technology of quantitative proteomics strategy. The reported data used state-of-the-art proteomics methodology generating extremely robust data. The manuscript in general is well written with an excellent description of the state of the art object of the research study. The methodologies and results and discussion were written in a very clear, concise manner and supported by relevant current literature to the study carried out.
Author Response
We've revised the article again, thanks!

Reviewer 2 Report
This paper is mainly on the study of the effects of different storage (refrigerated and frozen storage) on the quality of the shimp Litopenaeus vannamei. This quality was evaluated by physicochemical techniques, and the changes in muscle proteins by quantitative protemic technique to identify potention biochemical markers. quality degradation mechanisms. These will be very helpul in the future to understand the different mechanisms implicated in muscle quality deterioration.
This study was carried out with great care. Not being an English speaker myself, I am not in a position to judge whether the article is well written, or whether certain formulations need to be revised. Nevertheless I’ve some specific comments to point out:
- The title of your paper is not clear: this is more something like that: Use of TMT quantitative proteomics combined with physicochemical characteristics to identify potential biomarkers to follow the effects of cold and frozen storage of muscle tissue of Litopenaeus vannami.
- Don't forget to italicize the scientific names of species throughout your manuscript. Some are missing (lines 55, 69, 77 ….)
- Line 43: “dry consumption” isn’t clear here and it isn’t, to my opinion, correct.
- Lines 45 to 47: I think that you have to reformulate your sentence. These is the size and the number of crystals which are closely dependant on the temperature of storage, not the inverse.
- Line 66: …. Different types of microplastics (MPs) INDUCE (not have)
- Line 9_: Scientific
- Lines 108-109: for a better comprehension, please, erase “for low-temperature storage”
- Line 112: it’s a freezer, not a refrigerator.
- Lines 117-118: something wrong with the reference.
- Lines 142 to 149 are identical to 133 to 140 lines.
- Line 154: which is CAN?
- Line 177: 1e6 … what is that?
- Lines 179-180: what are MS2 and HCD?
- Lines 196 to 198: please erase the parentheses
- Line 241: more water is FREE not freed.
- Line 285: Table 2, not Table 3
- Line 313 or others: please refer the results you explain to the Table3
- Line 358: IN this study: you forgot “In”
- Line 363: Please, could you rewrite your sentence. The ice crystal formation is not caused by the water-holding capacity, this is the inverse.
- From line 390: there is no need to number the references as they have to be alphabetically ordered in your list.
Author Response
We have revised the review comments. Thank you.
